# How Animals Dance (When You're Not Looking)

## Abstract

We present a framework for generating music-synchronized, choreography aware animal dance videos. Our framework introduces choreography patterns—structured sequences of motion beats that define the long-range structure of a dance—as a novel high-level control signal for dance video generation. These patterns can be automatically estimated from human dance videos. Starting from a few keyframes representing distinct animal poses, generated via text-to-image prompting or GPT-4o, we formulate dance synthesis as a graph optimization problem that seeks the optimal keyframe structure to satisfy a specified choreography pattern of beats. We also introduce an approach for mirrored pose image generation, essential for capturing symmetry in dance. In-between frames are synthesized using an video diffusion model. With as few as six input keyframes, our method can produce up to 30 seconds dance videos across a wide range of animals and music tracks.

## 1 Introduction

> Everything in the universe has a rhythm; everything dances.
>
> —*Maya Angelou*

Humans dance spontaneously to music—just picture a toddler cheerfully bouncing to the beats at a birthday party. Animals can dance too; *Snowball the cockatoo*—can perform up to 14 distinct dance movements in response to different musical cues (Keehn et al., 2019). In fact, our animal friends are probably dancing all the time when we're not looking. In this paper, we capture this hidden world of animal dance, and expose it *for the first time* to the human eyes.

While we happen to be particularly obsessed with dancing animals, this paper introduces a new framework for generating music-synchronized, highly structured, up to 30 seconds long dance videos. Such capabilities are challenging for current state of the art generative models (Blattmann et al., 2023; Bar-Tal et al., 2024; Zeng et al., 2024; Yang et al., 2024), most of which are limited to short clips of a few seconds, do not produced synchronized audio and video, and lack intuitive controls for long range motion. Beyond text prompting, most controls for video generation are fine-grained and operate on a single frame at a time (Wang et al., 2023; Geng et al., 2024), *e.g.*, body pose, camera pose, motion brushes, etc. In contrast, we introduce *choreography patterns* as a new control for video generation. Specifically, we allow the user to specify a structured sequence of dance moves, or "beats", *e.g.*, A-B-A-B-C-D-A, where each letter corresponds to a particular move, and constrain the motion in the video to follow that choreography. Furthermore, we show how these choreography patterns can be automatically estimated from existing (human) dance videos.

Our use of choreography patterns as a control is inspired by how real dances are organized. A well-formed dance follows basic choreographic rules (Chen et al., 2021), which structure the movements to align with the rhythmic flow of the accompanying music, and often involve recurring patterns such as mirroring and repetition to help reinforce the musical structure (Kim et al., 2003; 2006). We leverage this inherent structure of dance to make the generation task more tractable. As input, we use a collection of initially generated keyframes, each representing a distinct pose. We then formulate the dance synthesis as a graph optimization problem, *i.e.*, find the optimal walk path through the keyframes where the underlying motion satisfy a specified choreography pattern of beats. Each selected keyframe in the path is aligned to a musical beat. The final dance video

is produced by synthesizing in-between frames between the keyframes using a generative video inbetweening model (Wang et al., 2024b;a) (Fig. 1).

Beyond 1) introducing a new type of generative video control (choreography patterns), and 2) a practical system for generating music-synchronized dance videos, this paper makes the following technical contributions. First, we introduce a technique for inferring choreography patterns from human dance videos, such as those found on Youtube and TikTok. Second, we formulate the satisfaction of these constraints as a graph optimization problem and solve it. Finally, we demonstrate an approach for pose-mirroring in the image domain, while retaining asymmetries in foreground and background features.

We demonstrate the effectiveness of our method by generating dance videos up to 30 seconds long across approximately 25 animal instances across 10 classes—including marmots, sea otters, hedgehogs, and cats—paired with various songs. These videos represent the first-ever recorded demonstrations of these animals performing such complex musical dance routines and will be studied by generations of zoologists.

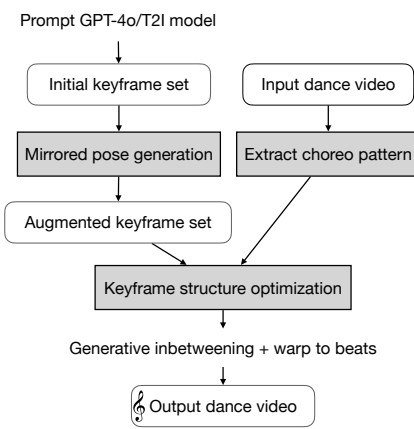

Figure 1: **System overview.** Given a few initially generated keyframes as input, we generate mirrored counterparts, extract choreography pattern from a dance video, and optimize the keyframe structure accordingly. The final dance is synthesized by generating in-between frames with a video diffusion model and warped to the musical beats. Our method is highlighted in gray.

## 2 PRIOR WORK

**Music-driven generative dance synthesis.** Earlier learning-based approaches developed neural networks that synthesize human dance motion directly from music input (Lee et al., 2018; 2019; Huang et al., 2020; Li et al., 2021; Zhang et al., 2022; Sun et al., 2020). Recent advances have shifted toward diffusion-based methods Qi et al. (2023); Le et al. (2023); Tseng et al. (2023), which also focus primarily on generating skeletal motions from music. More recently, some works have begun exploring direct dance video generation using video diffusion models (Sun et al., 2020; Ruan et al., 2023; Hong et al., 2025). However, directly enforcing choreography structure within these frameworks remains challenging. In addition, these learning-based approaches typically require training data—an issue in our case, as dance videos featuring animals are extremely scarce.

**Graph-based human dance motion synthesis.** In contrast to learning-based approaches, graph-based frameworks (Kim et al., 2003; 2006; Ofli et al., 2008; Manfrè et al., 2016; Chen et al., 2021) synthesize new motions from an existing dance motion segments database, and cast the dance synthesis as a graph optimization problem: finding an optimal path in the constructed motion graph that aligns with the input music. For example, Kim et al. (Kim et al., 2003) introduced rhythmic and beat-based constraints to guide the path search, while more recent work *ChoreoMaster* (Chen et al., 2021) incorporated richer choreography rules, requiring not only structural alignment with music but also stylistic compatibility.

Our approach follows this paradigm but differs in key ways. First, instead of relying on a motion capture database, we take a small set of keyframes of an animal or subject as input. We augment this set by generating mirrored pose images, creating a complete keyframe set for dance synthesis. The graph is constructed over these keyframes, and a video diffusion model is applied to generate realistic in-between frames along the optimized walk path, producing the final dance video. Second, while basic choreography rules can be inferred from the musical structure as done in (Chen et al., 2021), different performers may interpret the same piece differently. As such, we propose a way to extract choreography patterns directly from a reference dance video and use it as the control.

**Anthropomorphic character animation.** Given a reference image, character image animation, generates videos following a per-frame target human skeletal pose sequence. While existing methods (Hu, 2024; Hu et al., 2025) are primarily designed for human figures, recent work (Tan et al., 2024) extends to anthropomorphic characters by learning generalized motion representation. How-

ever, it still favor human-like anatomy, often producing animals with features like elongated limbs and human-style body proportions. They also struggle to generate high-fidelity videos from a single image when handling long and diverse sequences, such as a 30-second dance. In contrast, our method does not rely on per-frame human skeleton pose as guidance and uses choreography pattern as higher-level control, letting the video diffusion model generate in-between motions so that the final dance follows the choreographic structure, not human motion itself.

## 3 APPROACH

We begin by generating a small set of keyframes $\{I_k\}_{k=0}^{K-1}$, each depicting the subject, *e.g.*, a marmot, in different poses while maintaining a consistent background and static camera view (see Section 4.1 for details). Our goal is to synthesize a dance video of the input animal from the provided keyframes, synchronized to the beats and following the choreography pattern extracted from a reference dance video.

We first introduce how we extract the choreography pattern directly from a human dance video in Section 3.1. Since motion mirroring is an essential component of dance, we then present our approach for generating a mirrored pose counterpart for each keyframe to augment the keyframe set in Section 3.2. Finally, in Section 3.3, we present how to synthesize the full dance video using the complete set of keyframes, including mirrored ones, to follow the choreography pattern.

### 3.1 CHOREOGRAPHY PATTERN LABELING

Choreography are closely tied to the rhythmic structure of music. In music theory, a *beat* is the basic temporal unit, while a *bar* (or measure) groups a fixed number of beats. The *meter* defines how beats are grouped and emphasized within the bar, and is indicated by a time signature, *e.g.*, 2/4, 3/4, 4/4, where the upper number specifies beats per bar, and the lower number denotes the note value that receives one beat. In this work, we focus on music with a 4/4 time signature—each bar contains four quarter note beats—the most common structure in popular music.

**Problem definition.** Given a 4/4 music track with a synchronized dance video, we begin by detecting the beat times $\mathcal{B} = \{t_0, t_1, ..., t_{N-1}\}$, assuming a total of $N$ beats, corresponding to $\frac{N}{4}$ bars. Based on the beat times, we construct a sequence of motion segments $\mathcal{S} = \{s_0, s_1, ..., s_{\frac{N}{2}-1}\}$ where each segment $s_i$ spans from beat $t_{2i}$ to beat $t_{2i+1}$. Each bar thus yields two motion segments: one from the first to the second beat, and another from the third to the fourth beat, aligning with the 4/4 music structure where major movements typically begin on accented beats and end on weak ones, whereas transitions occur across weak-to-accented intervals. The "choreo pattern" labeling task outputs a sequence of labels $\mathcal{L} = \{l_0, l_1, ..., l_{\frac{N}{2}-1}\}$, *e.g.*, A-A'-B-C-D-D, where each $l_i$ corresponds to motion segment $s_i$. Distinct motions receive different labels, identical motions share the same one, and mirrored motions are indicated by prime-labeled counterparts (*e.g.*, A and A').

**Motion segments quantization.** We formulate motion segment sequence labeling as a quantization problem: clustering similar motion segments and assigning each a cluster ID as its label. Each segment $s_i$ of length $T_i$ is represented by the SMPL-X (Pavlakos et al., 2019) pose sequence recovered from the video: $s_i = \{(\theta_{t_i} \in \mathbb{R}^{3 \times (J+1)}, \tau_{t_i} \in \mathbb{R}^3)\}_{t_i=0}^{T_i}$, where $\theta_{t_i}$ contains per-joint axis-angle rotation for $J = 21$ body joints in addition to a joint for global rotation (the 0-th joint), and $\tau_{t_i}$ denotes the global translation in 3D space.

For clustering, we focus solely on poses—ignoring global translations—to capture distinctive motion patterns. The distance between two SMPL-X poses is defined as the average geodesic distance across joints:

$$d_\theta(\theta_1, \theta_2) = \frac{1}{J} \sum_{j=0}^{J} || \log(R(\theta_1^j)^T R(\theta_2^j)) ||_F \tag{1}$$

where $R(\theta^j)$ converts the axis-angle representation of joint $j$ into a rotation matrix. To account for slight temporal offsets between beats, we compute the distance between two motion segments using dynamic time warping (DTW), with $d_\theta$ as the local cost metric between poses. The clustering

function $\mathcal{C}$ is then defined as:

$$\mathcal{C}(\mathcal{S}; \mathrm{DTW}_{d_\theta}, \epsilon_\theta) \rightarrow \{\mathcal{C}_1, \mathcal{C}_2, ..., \mathcal{C}_C\} \tag{2}$$

where $\bigcup_{c=1}^{C} \mathcal{C}_c = \mathcal{S}$ , with $\mathcal{C}_i \bigcap \mathcal{C}_j = \emptyset$, for $i \neq j$. Segments within the same cluster satisfy: $\mathrm{DTW}_{d_\theta}(s_i, s_j) < \epsilon_\theta, \ \forall s_i, s_j \in \mathcal{C}_c$.

**Mirrored motion segments detection.** After the quantization stage, we identify mirrored motion segments in two steps.

*Mirrored pose clusters.* A mirrored joint rotation is defined by reflecting the axis-angle vector across the sagittal (YZ) plane: $\mathcal{F}(\theta^j) = (\omega_x, -\omega_y, -\omega_z)$, where $\theta^j = (\omega_x, \omega_y, \omega_z)$. Then a mirrored pose $\theta'$ is obtained by applying this reflection to each joint after left-right joint swapping:

$$\theta'^j = \mathcal{F}(\theta^{\pi(j)}) \quad \text{for } j = 0, \dots, J \tag{3}$$

here $\pi(j)$ denotes the left-right joint permutation (*e.g.* swapping left/right arms, legs, and shoulders), and for central joints (*e.g.*, spine, neck, head), $\pi(j) = j$, so only the reflection is applied.

Two clusters $\mathcal{C}_a$ and $\mathcal{C}_b$ are considered mirrored if there exists at least one pair of segments $s_i \in \mathcal{C}_a, s_j \in \mathcal{C}_b$, such that the mirrored version of $s_i$, denoted by $s'_i = \{\theta'_{t_i}\}_{t_i=0}^{T_i}$, is similar to $s_j$ under dynamic time warping: $\mathrm{DTW}_{d_\theta}(s'_i, s_j) < \epsilon_{\theta'}$. The resulting set of mirrored cluster pairs is: $\mathcal{M} = \{(\mathcal{C}_a, \mathcal{C}_b) | \ \exists s_i \in \mathcal{C}_a, s_j \in \mathcal{C}_b, \mathrm{DTW}_{d_\theta}(s'_i, s_j) < \epsilon_{\theta'}\}$.

*Mirrored motion directions within a cluster.* For clusters without a mirrored counterpart, we further check whether they can be internally partitioned into two directionally mirrored groups. We first extract the overall motion direction $\vec{d}_{s_i}$ of each motion segment $s_i$ using its global translation: $\vec{d}_{s_i} = (\tau_{t_i}^{T_i} - \tau_{t_i}^0)/||\tau_{t_i}^{T_i} - \tau_{t_i}^0||$, where $\tau_{t_i}^0$ and $\tau_{t_i}^{T_i}$ denote the segment's start and end positions, respectively.

To identify mirrored directions, each motion direction $\vec{d}_{s_i}$ is reflected across the YZ plane as $\vec{d'}_{s_i} = \mathrm{diag}([-1, 1, 1])\vec{d}_{s_i}$. We then perform bipartite matching to find mirror pairs $(s_i, s_j)$ that satisfy $||\vec{d'}_{s_i} - \vec{d}_{s_j}|| < \epsilon_d$. If valid pairs are found, we assign all matched segments into two directionally consistent groups based on their directional similarity. The original cluster $\mathcal{C}_i$ is then partitioned into two mirrored subgroups $(\mathcal{C}_i^0, \mathcal{C}_i^1)$, which are then added to the mirrored cluster set $\mathcal{M}$.

Finally, we assign a unique label to each cluster. For each mirrored pair $(\mathcal{C}_a, \mathcal{C}_b) \in \mathcal{M}$, we assign a base label $l_a$ (*e.g.* A) to segments in $\mathcal{C}_a$, and its mirrored label $l'_a$ (*e.g.* A') to segments in $\mathcal{C}_b$. Clusters without a mirrored counterpart are assigned a distinct label without a prime.

## 3.2 MIRRORED POSE IMAGE GENERATION

To augment the keyframe set with mirrored counterparts, we generate visually consistent keyframe pairs for each input pose. This process (Fig. 2) involves fine-tuning a text-to-image model with ControlNet, generating mirrored edge maps, and re-generating the original keyframes for consistency. In the end, we get a complete set of consistent keyframes $\mathcal{I} = \{I_0, \dots, I_{K-1}, I'_0, \dots, I'_{K-1}\}$, where $I'_k$ is the mirrored version of $I_k$.

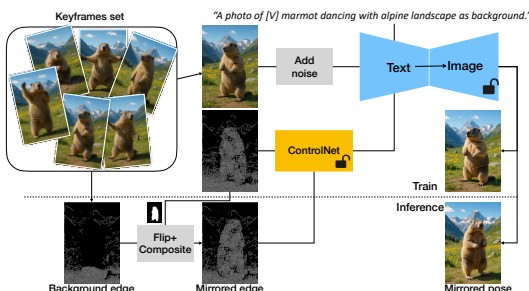

Figure 2: **Mirrored pose generation.** We fine-tune a text-to-image model with ControlNet using the canny edges extracted from each keyframe as conditioning. During inference, mirrored pose images are generated by flipping only the subject edges and using an inpainted background edge composed from the keyframe set.

**Fine-tuning.** We fine-tune a pretrained text-to-image model on the input keyframes set, over-fitting it to capture the appearance of the specific input subject instance and the background. To provide structural guidance, we incorporate ControlNet (Zhang et al., 2023) using the canny edge maps extracted from the input images as a conditional input. We use the prompt format: "A photo of [V] [*subject class name*] dancing [*background description*].", where [V] is a unique token for identifying the subject **instance** rather than **class**. The placeholders [*subject class name*]

and [*background description*] are replaced with the actual class name of the subject and background description. For example, "A photo of [V] marmot dancing with alpine landscape as background."

**Mirrored edge generation.** To generate mirrored pose images, We first extract the subject mask using SAM (Kirillov et al., 2023). We then construct a unified background canny edge map by inpainting and stitching the background edges from all input keyframes. For each keyframe, we extract the subject's edge map and horizontally flip it to create a mirrored subject edge map. This flipped edge map is then composited with the shared background edge map to generate a full mirrored edge map. which is used as input to the fine-tuned model to generate the corresponding mirrored image.

**Keyframes re-generation.**

The original keyframes may contain slight inconsistencies in the background due to generation instability. Additionally, the fine-tuned model may introduce subtle color shifts during inference. To ensure visual consistency among the augmented keyframes set, we regenerate the original keyframes using the same model and shared background edge map (see Fig. 3 for an example).

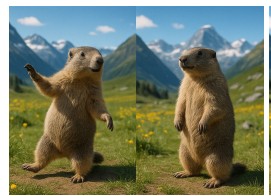 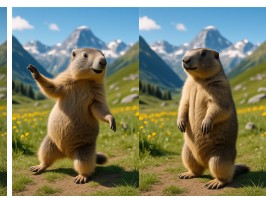

(a) raw keyframes      (b) refined keyframes

Figure 3: Improving visual consistency by regenerating keyframes with shared background edges.

**Implementation details.** We use FLUX.1-dev and Xlabs-AI/flux-controlnet-canny as the pretrained text-to-image and controlnet model. We fine-tune them jointly with a LoRA rank of 16 for 500 epochs, Training on 6 keyframes takes around 90 minutes on a single A100 GPU. Canny edges are extracted using threshold values of (50, 100).

### 3.3 CHOREOGRAPHY PATTERN DRIVEN DANCE SYNTHESIS

Given the augmented keyframe set $\mathcal{I} = \{I_0, \ldots, I_{K-1}, I'_0, \ldots, I'_{K-1}\}$, where $I'_k$ denotes the mirrored counterpart of keyframe $I_k$, and the choreography pattern label sequence $\mathcal{L} = \{l_0, l_1, ..., l_{\frac{N}{2}-1}\}$, the goal is to find an optimal walk path $\mathcal{P} = \{I_{p_0}, I_{p_1}, \ldots, I_{p_{N-1}}\}$ through the keyframe set, and the $i$-th keyframe $I_{p_i}$ in the path corresponds to the $i$-th beat. We then apply video diffusion model to generate in-between frames, finally producing the final dance video.

Since each label $l_i$ corresponds to a motion segment between keyframe pairs $(I_{p_{2i}}, I_{p_{2i+1}})$, we cast path planning as a graph optimization, where each node represents a candidate keyframe pair. The choreography label sequence $\mathcal{L}$ specifies assignment constraints: same labels map to the same pair, distinct labels to distinct pairs, and mirrored labels to mirrored pairs. The object is to assign each label $l_i$ to a node such that these constraints are met while minimizing the total transition cost along the path.

**Keyframe graph construction.** We construct the keyframe graph $G = (V, E)$ as a directed graph, where each node $(I_u, I_v) \in V$, with $I_u \neq I_v$, represents an ordered pair of keyframes from the augmented keyframe set $\mathcal{I}$. Note $(I_u, I_v) \neq (I_v, I_u)$. Each node corresponds to a potential dance segment from $I_u$ to $I_v$, and each edge $(I_u, I_v) \rightarrow (I_w, I_x) \in E$, with $I_v \neq I_w$, represents a valid transition between segments.

To ensure both expressive motion and synthesis feasibility, we filter nodes based on the average flow magnitude $|F(I_u, I_v)|$ from $I_u$ to $I_v$, computed over the foreground region of the subject. We use RAFT (Teed & Deng, 2020) to compute the optical flow. Nodes with flow that is too small or too large are discarded. Only node pairs with acceptable motion range are retained:

$$V = \{(I_u, I_v) \mid M_{\text{low}} < |F(I_u, I_v)| < M_{\text{high}}\} \tag{4}$$

To make the synthesized dance smooth and fluid, we define the edge cost between two nodes as the flow magnitude between the end keyframe of the first node and start one of the next node: $\mathcal{T}((I_u, I_v) \rightarrow (I_w, I_x)) = |F(I_v, I_w)|$. We prune high-cost transitions by including only edges with flow below a threshold:

$$E = \{((I_u, I_v) \rightarrow (I_w, I_x)) \mid |F(I_v, I_w)| < M_{\text{high}}\} \tag{5}$$

We also define a mirroring function $\mu : V \rightarrow V$ over graph node such that two nodes are mirrored if and only their respective keyframes are mirrored: $\mu((I_u, I_v)) = (I'_u, I'_v)$.

**Graph optimization.** We define a node assignment function: $\phi : \mathcal{L} \rightarrow V$, which maps each choreography label $l_i \in \mathcal{L}$ to a graph node $(I_u, I_v) \in V$, forming a walk path through the keyframe graph. The goal is to find the optimal assignment $\phi^*$ that minimizes the total transition cost across the sequence:

$$\phi^* = \text{argmin} \sum_{i=0}^{\frac{N}{2}-2} \mathcal{T}(\phi(l_i), \phi(l_{i+1}))$$ (6)

subject to the following constraints:

$$l_i = l_j \ \Leftrightarrow \ \phi(l_i) = \phi(l_j), \ l'_i = l_j \ \Leftrightarrow \ \phi(l_i) = \mu(\phi(l_j))$$ (7)

To ensure visual variety and avoid redundancy, we introduce two additional constraints:

(1) Different labels cannot map to partially mirrored node pairs—defined as nodes sharing one keyframe (in any order), with the other keyframes mirrored:

$$l_i \neq l_j, \Rightarrow (I_a, I_b) \not\approx (I_c, I_d)$$ (8)

where $\phi(l_i) = (I_a, I_b), \phi(l_j) = (I_c, I_d)$.

(2) Consecutive, distinct, and non-mirrored labels must not be assigned to nodes that share a keyframe, to prevent unnecessary single keyframe repetition.

$$l_i \neq l_{i+1}, \ l'_i \neq l_{i+1} \ \Rightarrow \ (I_a \neq I_c \wedge I_b \neq I_d)$$ (9)

where $\phi(l_i) = (I_a, I_b), \phi(l_{i+1}) = (I_c, I_d)$.

### 3.4 WARP TO MUSIC

We generate the final dance video by applying a video diffusion model to synthesize in-between frames along the optimized keyframe walk path $\mathcal{P} = \{I_{p_0}, I_{p_1}, \ldots, I_{p_{N-1}}\}$, where each keyframe $I_{p_i}$ corresponds to beat position $i$. Note that since there are motion repetition in the choreography, we only have to synthesize videos between unique keyframe pairs. In practice, we use Framer (Wang et al., 2024a), which generates 14 in-between frames, and we assume a fps of 25. To synchronize with the music, we warp the video timeline such that the timing of every keyframe in $\mathcal{P}$ align with the corresponding beat time in the audio. Following the visual rhythm strategy from (Davis & Agrawala, 2018), we accelerate the warping rate into beat points and decelerate before and after to preserve beat saliency while ensuring temporal smoothness.

## 4 EXPERIMENTS

We generate a collection of input keyframe grids featuring approximately 25 animal instances, some captured as half-body views. The animal classes include *marmot, capybara, hedgehog, meerkat, penguin, sea otter, cat, quokka, beaver* and others. We also incorporate characters such as *Elmo*. For the video results, we generate dance videos for these instances using five popular song clips, ranging from 16 to 28 seconds in length, with choreography patterns extracted from the corresponding YouTube video clips. While we showcase our method using these examples, it can adapt to any choreography patterns paired with music. Fig. 6 shows selected examples of the final keyframe pairs assigned for each choreography label, arranged in the order specified by the choreography pattern. We highly recommend viewing the supplementary video for the full experience.

### 4.1 KEYFRAMES GENERATION

We generate initial keyframes set by prompting text-to-image model FLUX to generate an image grid with consistent keyframes using prompt template like "a 3x2 grid of frames, showing 6 consecutive frames from a video clip of [...]". For example, the description prompt might be *"a marmot dancing wildly in the wild alpine landscape, striking a variety of fun and energetic poses"*. The same

prompt format can also be used with GPT-4o, which supports even finer specifications. For instance, one can specify: *"Generate a grid with 2 rows and 3 columns. Depict a quokka with distinct poses with a wild background. The postures should feel natural for the quokka's anatomy...."*. In all of our results, we use a total of 6 input keyframes.

## 4.2 CHOREOGRAPHY PATTERN LABELING

We collect a total of 20 dance video clips featuring various 4/4 music tracks from Youtube and TikTok, ranging from 12s to 28s in length. To create ground truth, we manually annotate the choreography pattern label sequence. We then evaluate our method in Section 3.1 by comparing our extracted label sequences against the ground truth ones. Beat times

| Clustering | | Mirroring | | |
|---|---|---|---|---|
| ARI↑ | NMI↑ | Prec. ↑ | Recall↑ | F1↑ |
| 0.94 | 0.98 | 0.93 | 0.91 | 0.92 |

Table 1: Evaluation on choreography pattern labeling.

are detected using Librosa, and SMPL-X pose sequences are recovered from the videos using GVHMR (Shen et al., 2024). We set the threshold values as follows: $\epsilon_\theta = 0.21, \epsilon_{\theta'} = 0.25$ and $\epsilon_d = 0.1$. Specifically, we evaluate two aspects: (1) Clustering accuracy, where each unique label—including mirrored variants—is treated as a distinct cluster. We assess the clustering results using standard metrics: Adjusted Rand Index (ARI) and Normalized Mutual Information (NMI); (2) Mirror detection accuracy, where we compute precision and recall based on the correctly identified mirrored motion segments pairs.

We report the results in Table 1. Our training-free extraction method achieves overall strong quantization accuracy, effectively differentiating different motion patterns. For mirroring, our prediction occasionally misses mirrored pairs, typically in cases where the poses are symmetrical but exhibit subtle mirroring in head or body orientation. Since our method also can output representative motions for each choreography label, users can more easily visualize the structure and manually correct annotation errors if needed.

## 4.3 BASELINE COMPARISONS

**Baselines.** A straightforward baseline is music-conditioned video generation, which generates videos directly from audio and text prompts. However, methods such as MusicInfuser (Hong et al., 2025) are trained on human dance videos and typically produce only short clips, *e.g.*, 5s. So they cannot generalize to long animal dance videos synchronized to the input music. Several examples are provided in the supplementary videos.

Next we compare our method to human pose driven single image animation method using Animate-X (Tan et al., 2024), which animates an input image according to a sequence of human skeleton poses and works with anthromorphic characters. For each of our generated dance videos, we extract the pose sequence from the same reference video used to extract the choreography pattern and use it as the driving sequence for Animate-X.

**User study.** Since there are no existing long, structured animal dance videos for direct reference, we conducted a perceptual user study to evaluate the generated dance videos. We used 40 generated dance video pairs across 5 different songs, and invited 31 participants. Each participant was presented a random set of 8 pairwise comparisons of our results and

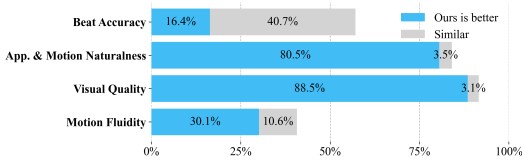

Figure 4: User ratings of our approach compared to Animate-X on various criteria.

Animate-X. The order of the videos within each pair was randomized. For each pair, participants were asked to choose which video they judged better on each of four criteria, or select "similar" if they found no difference: (1) *Beat accuracy*—are the dances synchronized with the music beats? (2) *Appearance & motion naturalness*—do the animals' body proportions and movements feel natural for them? (3) *Visual quality*—do the videos have high overall visual quality (*e.g.* sharpness, clarity, and fewer artifacts)? (4) *Motion fluidity*—are the dance movements smooth and fluid? The responses are shown in Fig. 4, and example qualitative comparisons are shown in Fig.5.

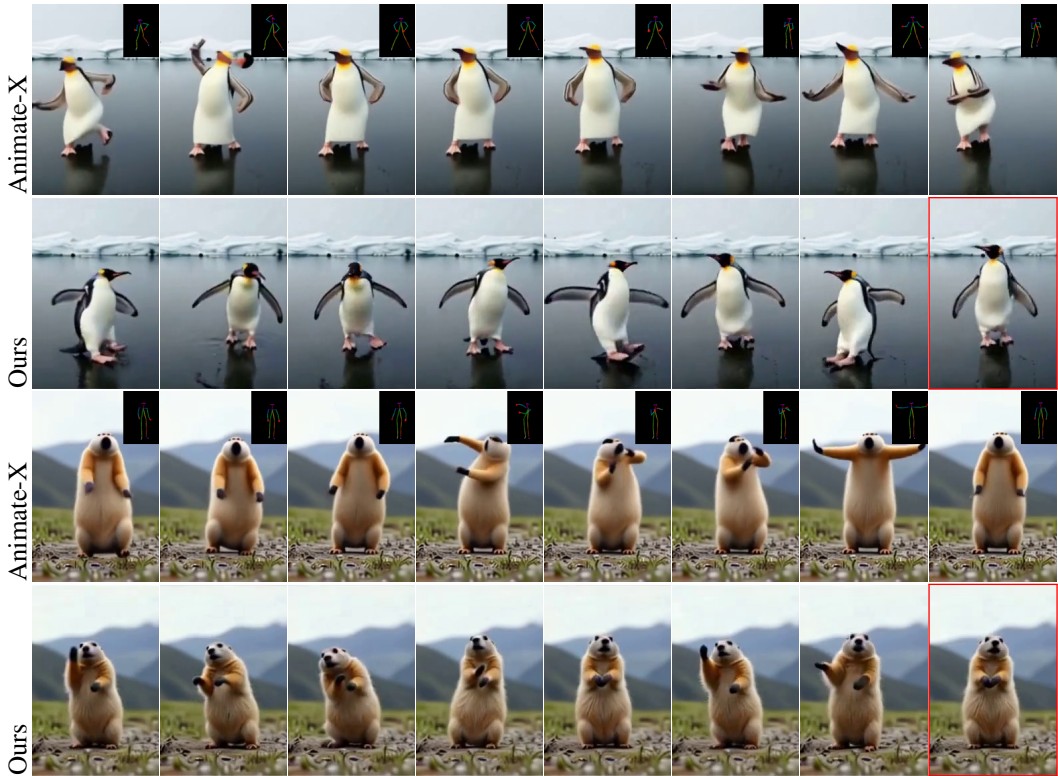

Figure 5: Sample frames from both our results (cropped for visualization) and Animate-X. Top: *APT* (Youtube [DJz1zlm73HI]); Bottom: *Cann't Stop Feeling* (Youtube [xyMBnn3dzdU]) used as the reference dance videos. The red box marks the input image for Animate-X. See supplementary for the full video comparison.

**Discussions.** While beat accuracy was rated similarly for both methods, participants found our results more natural-looking for animals and of higher visual fidelity than those from Animate-X. Specifically, $80.5\%$ of responses rated our videos superior in appearance and motion naturalness, and $88.5\%$ rated them higher in overall visual quality. Animate-X was preferred for motion fluidity ($59.3\%$).

These results are in line with the different setups of these two methods. Animate-X generates animal dances by following fine-grained per-frame human pose sequences, which naturally leads to human-like figures and dance motions—resulting in more fluid and richer movement compared to our method. Yet transferring human poses to animal bodies is inherently difficult: it requires to solve complex correspondence across different body morphologies, and becomes even more challenging when handling the long and diverse pose sequences of real dances. For example, in Fig. 5, Animate-X maps the human arms to the penguin's wings, causing the wings to move like human arms; the penguin's differently shaped head further introduces blurry artifacts. Our method instead uses choreography pattern as higher-level control, letting the video diffusion model generate in-between motions so that the final dance follows the intended dance structure rather than fined-grained poses.

## 5 DISCUSSION & LIMITATIONS

We present a paradigm for generating music-synchronized, choreography-aware animal dance videos by using choreography pattern as a novel control to impose a structure on the keyframes input. Our work opens up exciting opportunities for creative and fun applications of dancing animals in entertainment and social media. Below we discuss limitations and future works.

**Limitations.** We use an offline video diffusion model to generate short motion segments between keyframes (*e.g.*, 0.5s for a 120 BPM song). The motion can sometimes look unrealistic: animals

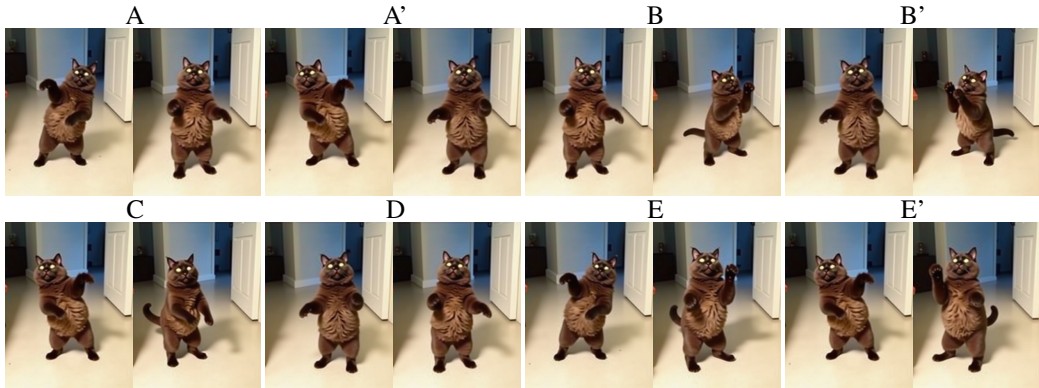

(a) Dance to *Uptown Funk* following the choreography pattern extracted from Youtube [U9Zj1BaH01c] (16.0s to 36.0s): A-A'-A-A'-A-A'-A-A'-B-B'-C-D-E-E'-E-E'.

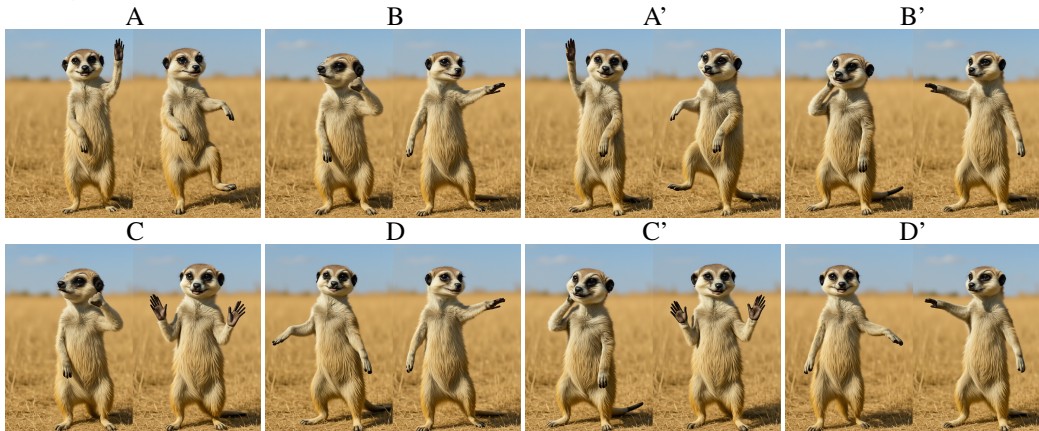

(b) Dance to *Bumblebee* following the choreography pattern extracted from Youtube [GIq7ZgmxE2w] (15.5s to 45.5s): A-B-A'-B'-C-D-C'-D'-A-B-A'-B'-C-D-C'-D'-E-F-G-G-G-G-H-H-H'-H'-G-G-G-G-H-H.

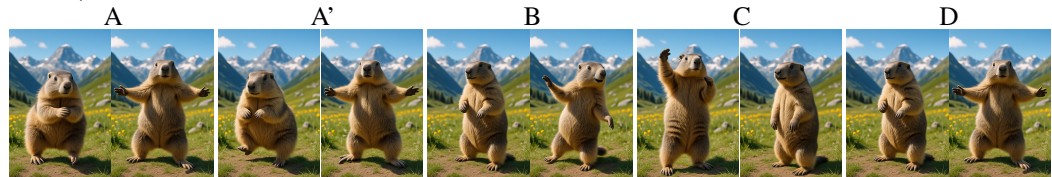

(c) Dance to *Rasputin* following the choreography pattern extracted from Youtube [jkRIIH42Vo8] (12.0s to 33.24s): A-A'-A-A'-A-A'-A-A'-B-B-B-B-B-B-B-B-C-C-C-C-D-D'-D.

Figure 6: Selected examples from our generated dances. Keyframe pairs are labeled by the choreography pattern label, arranged in the order specified by the choreography pattern. See supplementary for the full dance video with music.

may appear to slide or morph between poses rather than moving in a physically plausible way. This stems from the limitations of current video diffusion models in producing naturalistic motion for articulated subjects. However, we are optimistic that these issues can be addressed with continued advances in large-scale video diffusion models,

**Future works.** To generate more advanced and musically aligned animal dances, two directions can be explored: (1) *dance motion realism*: the motions generated by the video diffusion model may not always reflect plausible or expressive dance motion. Incorporating priors that favor natural, dance-like movement could improve alignment with musical context. (2) *style compatibility*: although our method follows the choreography pattern, it does not consider musical style. Modeling genre-specific movement characteristics could enhance the stylistic coherence of generated dances.

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

# A APPENDIX

## A.1 LLM USAGE STATEMENT

LLM was used to correct grammar and improve the fluency of the writing in this paper.

## A.2 USER CONTROL

Given the same choreography pattern for the dance, the user can use pose-grid template to guide the input keyframe poses, control the allowed motion range in the graph, and define custom constraints during graph optimization.

**Pose-grid template control.** Given a keyframe pose grid as template, we prompt GPT-4o to generate a new grid in which another animal "mimics" each pose from the original, though the poses are not expected to be exact same since different animals have different anatomical structures. This template may come from a previously generated grid (see Fig. 7 for an example) or be extracted from a human dance video by identifying distinct poses. This provides a way to guide or customize the input poses, which allows to generate dance videos where different animals dance alike (see supplementary video for examples).

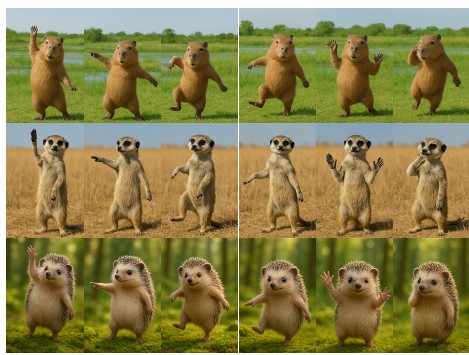

Figure 7: *Keyframe pose grid mimicking.* Top row: A keyframe pose grid template showing six distinct poses of a capybara. Middle and bottom rows: A meerkat and a hedgehog mimicking the capybara's poses, while preserving their own body structure, generated using GPT-4o.

**Motion range control.** The threshold parameters $M_{\text{low}}$ and $M_{\text{high}}$ control the range of allowed motion magnitudes between keyframes. A typical setting is $M_{\text{low}} \geq 12.0$ and $M_{\text{high}} \leq 60.0$ at resolution $1024 \times 576$. Within this range, lowering $M_{\text{low}}$ and increasing $M_{\text{high}}$ introduces more candidate nodes and transitions, potentially resulting in richer and more expressive dances.

**User custom constraints control.** During graph optimization, users can specify hard constraints on node assignments—for instance, enforcing preferred keyframe pair(s) for specific label(s). Additionally, for some dance, mirrored poses happen at the start and end of a choreography label. When such mirrored pose pairs are detected for a specific label, we can enforce corresponding node assignments during optimization. For example, such labels have to be assigned to nodes $(I_u, I_v)$ where $I_v = I'_u$, This allows closer alignment with the reference choreography.

## A.3 FAILURE CASES

**Motion intensity estimation.** Within the keyframe graph, we estimate the underlying motion strength between keyframe pairs using the average flow magnitude. This measure can be unreliable in certain cases. For example, when two keyframes depict mirrored side views, the flow fails to capture the true motion complexity between poses. In Fig. 8, the average flow magnitude is 36.81 (image size is $1024 \times 576$), which appears moderate due to incorrect correspondences between opposite sides, but the sea otter must rotate from one side view to the other, and this motion is more complex and sometimes

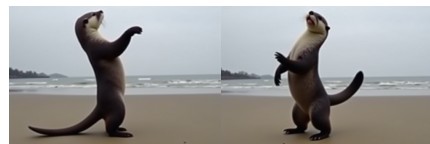

Figure 8: *Failures.* The average flow magnitude from left to the right is not large, but the underlying motion intensity is much higher.

challenging for the video model to synthesize. Establishing a reliable link between keyframe flow and the generated motion strength remains an open problem.

**Background consistency.** As described in Sec. 3.2, we re-generate the keyframes with the fine-tuned model to improve visual consistency in the initial keyframe set using a shared background

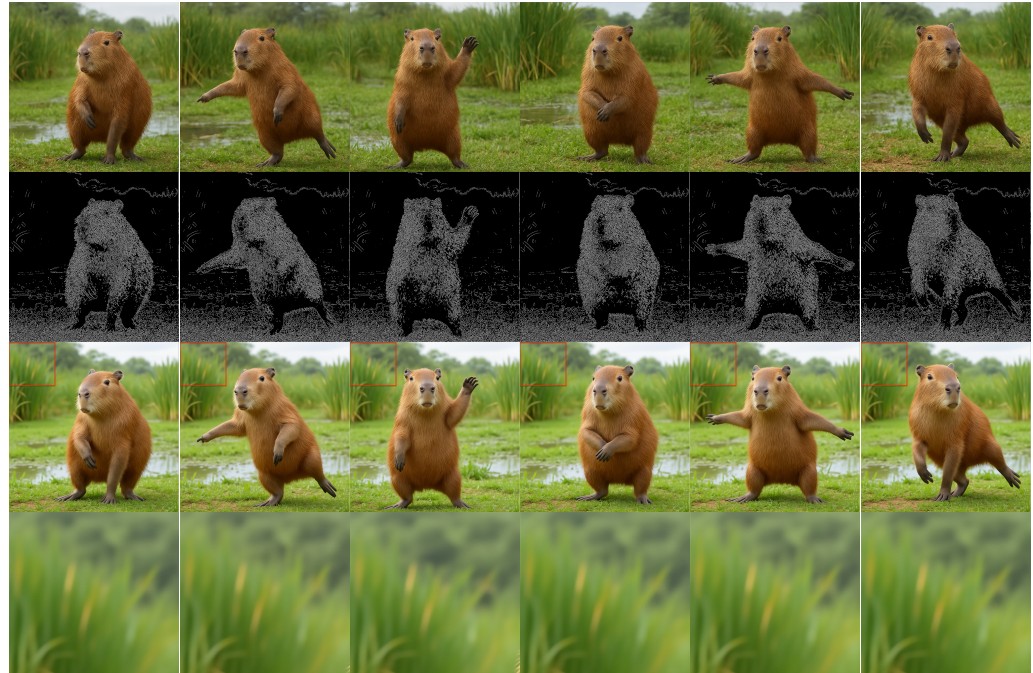

Figure 9: Background inconsistency caused by missing background Canny edges in the original keyframes. Top row: original keyframes generated by GPT-4o. Second row: Canny edge maps used in generating the refined keyframes. Third row: our refined keyframes. Bottom row: zoom-in of the top-left corners (red rectangles) of the refined keyframes, highlighting slight background inconsistencies where Canny edges are absent.

Canny edge map which is inpainted from the original keyframes as control. However, slight background consistency can remain when the original keyframes have shallow depth of field, which limits the Canny edge extraction. For example, in Fig. 9, the grass background behind the capybara is blurred in the original keyframe, leaving no detectable edges in that region. As a result, the refined keyframes show slight background inconsistencies in the grass (see bottom row). However, for scenes with clear and sharp backgrounds, our method maintains consistency well.

