# OpenReview forum: "How Animals Dance (When You're Not Looking)"
_ICLR.cc/2026/Conference — ICLR 2026 Conference Withdrawn Submission_

### Official Review · Reviewer_ssGY · 2025-10-31

**Soundness:** 3
**Presentation:** 3
**Contribution:** 2
**Rating:** 4
**Confidence:** 3

**Summary:**

In this paper, the authors propose a framework which 1) generate dance keyframes from text prompts for different animals,
2) extract music tempo and adjust the keyframes, 3) perform a video inbetweening generation for the final video.

We are seeing nice and cute videos where animals are dancing to the music.

**Strengths:**

1. I think it is overall a nicely written graphics paper with a clear and concise introduction.
The combination use of dynamic time warping, pose clustering, RAFT, graph optimization and
generative video models show that the authors have a in depth understanding of the graphics pipeline, traditional and newly emerging generative approaches.

2. The authors are honest about limitations and future work directions.
They discuss the failure cases (motion artifacts from video diffusion, background inconsistency) and future work directions.

3. A comprehensive set of user studies are conducted to evaluate the quality of the generated videos.

**Weaknesses:**

1. The motion / animation quality is not very ideal.
There're visually very noticeable artifacts and jumps in the motion.
While this might not be fair to compare indiviual researchers to industrial product teams which have far bigger budgets and resources,
I think in terms of the visual quality and the motion smoothness for music conditioned video generation,
this demo is not as good as the online entertainment applications released on social media platforms like TicTok.

2. The keyframe formulation does not feel intuitive or natural,
and the applications require a relatively static background and camera view,
which limits the level of freedom in entertainment of the proposed method.

3. The engineering pipeline is too long and thus making it unscalable.
A large number of components are added and a huge ton of hyperparameters are introduced,
with which it hard to tell how the proposed method can be generalized to other datasets and tasks outside this specific application.

4. There's relative limited contribution in terms of the novelty and the significance of the proposed method.
While this provides a solid engineering pipeline, it might not be most suitable for the submission of ICLR.

**Questions:**

1. how long does one generation takes;
Is there a break down of each module and human effort?

---

### Official Review · Reviewer_kGx5 · 2025-11-01

**Soundness:** 3
**Presentation:** 3
**Contribution:** 1
**Rating:** 4
**Confidence:** 4

**Summary:**

This paper presents a novel framework for generating music-synchronized animal dance videos using a small set of keyframes. The method leverages text-to-image models (or GPT-4o) to generate initial animal pose keyframes, then formulates dance synthesis as a graph optimization problem to align poses with rhythmic beats extracted from reference dance videos. To enhance visual coherence, a mirrored pose generation strategy is proposed for capturing bilateral symmetry in dance movements. The final continuous animation is produced using a video diffusion model for in-between frame synthesis. The system can produce up to 30-second dance clips from as few as six input keyframes.

**Strengths:**

* Combining animal motion generation, musical beat synchronization, and keyframe optimization is original and creative.
* The mirrored pose generation step addresses a real aesthetic issue in animal dancing (natural symmetry).

**Weaknesses:**

* The mirrored pose generation step makes the generation unreal.
* I have seen the demos in the website, and I find that the generated result looks too stiff and lacks detail.

**Questions:**

Compared with end-to-end approaches like [*Animate Anyone*](https://arxiv.org/pdf/2311.17117), I think focusing only on choreography patterns loses many important details. Can the current framework handle this issue?

---

### Official Review · Reviewer_hxcn · 2025-11-01

**Soundness:** 3
**Presentation:** 4
**Contribution:** 2
**Rating:** 4
**Confidence:** 4

**Summary:**

This work focuses on generating animal dance videos from music.
Given an input music track, the system outputs a video of an animal dancing in synchronization with the rhythm.

Unlike AnimateAnyone, this paper adopts a two-stage pipeline:
first generating keyframes, and then arranging dance units along the music beats via graph-based computation.
According to the paper, this design aims to achieve more stable image quality and to avoid the motion artifacts that can occur in models such as AnimateX.

However, from another perspective, the results seem quite limited in scope.
Most of the generated dances still resemble human-like movements performed by animals, rather than exhibiting species-specific motion patterns, such as tail dynamics or the non-limb-based movements of snakes, spiders, or fish.
This raises questions about the applicability and generalization capacity of the proposed framework.

**Strengths:**

1. The idea of first generating keyframes and then arranging dance units according to the music through a graph-based warping strategy is quite interesting. This design helps maintain stable image quality while producing coherent video sequences.

2. The writing is clear and easy to follow, and the presented demos are very cute and engaging.

3. The experimental results are comprehensive and well-presented.

**Weaknesses:**

1. From the results, the generated dances still appear limited to humanoid structures and behaviors. However, this paper claims to focus on animals. Why, then, does the framework rely on a human-based model (SMPL)? Can it be applied to non-humanoid species such as fish, snakes, reptiles (e.g., dinosaurs), or spiders?
This leads to a more fundamental question: where does this framework actually generalize?
If the intended motion domain remains largely human-like, why not directly adopt existing methods such as AnimateX, which are designed for articulated human motion?
The authors mention that AnimateX tends to produce artifacts for large motions. Then why not instead constrain the generation to smaller, more controlled music-to-motion segments?

2. From a novelty perspective, the idea of treating dance units as graph vertices and their connections as edges has already been explored in 3D choreography systems, such as ChoreoMaster [1].
What is fundamentally different about the proposed approach compared to those prior works, in terms of principle or formulation?

[1] Chen et al., SIGGRAPH 2021, ChoreoMaster

Generally, while the results are indeed cute, they are not particularly convincing me this is really useful. Similar effects, such as dancing animals or expressive pet animations, can already be achieved using tools like Animate Anyone.
The current demos do not convincingly demonstrate broader capability.
**If the authors could somehow make out unconventional examples (a dancing spider or fish) it would make the contribution much more compelling, and I would be happy to raise my score accordingly.**

**Questions:**

see above

---

### Official Review · Reviewer_a366 · 2025-11-01

**Soundness:** 3
**Presentation:** 3
**Contribution:** 2
**Rating:** 4
**Confidence:** 3

**Summary:**

This paper presents a novel framework for generating long-form, music-synchronized, and choreography-aware animal dance videos. The core innovation is the introduction of "choreography patterns" as a high-level control signal, which are automatically extracted from reference human dance videos. Starting with a small set of generated keyframes of an animal, the method formulates the dance synthesis as a graph optimization problem. The final video is rendered by synthesizing in-between frames using a video diffusion model. The authors demonstrate the ability to create dance videos up to 30 seconds long across various animals.

**Strengths:**

The paper presents a creative application of "dancing animals" and addresses a practical limitation of current models, which lack intuitive controls for creating coherent, long-duration motion.

Decomposing the complex task into a graph optimization problem is an effective strategy. It decouples the high-level temporal structure (the choreography) from the low-level frame generation (the diffusion model), making the overall problem more tractable.

The paper also includes a well-designed perceptual user study comparing the proposed method against a baseline (Animate-X), which provides valuable, quantifiable insights into the method's pros and cons.

**Weaknesses:**

1. The core innovation in "choreography patterns" for long-range control has been previously explored in the 3D skeleton animation domain (e.g., ChoreoMaster). The main contribution here appears to be a combination of "automated choreography extraction from video" (also a relatively established task) with "keyframe-based video generation", which lacks a fundamental methodological breakthrough.

2. The demo reveals less fluid generated motion compared to the baseline with a distinct "pose-to-pose" quality, which ultimately undermines the overall quality of the final video. Fig. 4 also reveals the trade-off between "appearance & motion naturalness" and "motion fluidity" compared to the baseline. This suggests the limitation of the keyframe-based graph approach than methods driven by per-frame guidance. Compared to the complex dance genres explored in the broader dance synthesis field (e.g., hip-pop, waltz, ballet), the complexity and artistic expressiveness of the generated dances are rudimentary. The framework's capability to handle more advanced and nuanced choreography remains unproven.

3. While direct baselines for animal dance generation are scarce, its core dance synthesis capability can still be evaluated against more sota methods in avatar dance. Concatenated baseline ("skeleton-driven animation + motion retargeting") is also available (e.g., Skinned Motion Retargeting). The absence of such a comparison weakens the paper's claims about its method's superiority.

4. The paper addresses a relatively narrow application scenario. Beyond creating amusing short videos for social media, the underlying technical motivation and broader utility are not well-articulated, making the research feel somewhat niche and constrained.

5. The paper's claim of generating videos up to 30 seconds long is potentially misleading. This is not achieved by a single model capable of understanding and generating long-term dynamics but an engineered composition with an independent video in-betweening model.

**Questions:**

Do you consider the lack of motion fluidity to be an inherent limitation of the graph-search-plus-in-betweening paradigm?

Is the framework able to handle more complex and less repetitive choreography, such as the rapid, consecutive movements or fine-grained body control?

How do you position your contribution in the context of true, coherent long-video generation? What are the advantages of your approach compared to works that attempt to model long-term dependencies within a single, unified model?

---

### Note · Authors · 2025-11-13

I have read and agree with the venue's withdrawal policy on behalf of myself and my co-authors.